# Delta Variant of SARS-CoV-2 Replacement in Brazil: A National Epidemiologic Surveillance Program

**DOI:** 10.3390/v14050847

**Published:** 2022-04-20

**Authors:** Joice P. Silva, Aline B. de Lima, Luige B. Alvim, Frederico S. V. Malta, Cristiane P. T. B. Mendonça, Paula L. C. Fonseca, Filipe R. R. Moreira, Daniel C. Queiroz, Jorge G. G. Ferreira, Alessandro C. S. Ferreira, Renan P. Souza, Renato S. Aguiar, Danielle A. G. Zauli

**Affiliations:** 1Departamento de Pesquisa & Desenvolvimento, Instituto Hermes Pardini, Belo Horizonte 31270-901, Brazil; joice.silva@grupopardini.com.br (J.P.S.); alinebrito.lima@grupopardini.com.br (A.B.d.L.); luige.alvim@grupopardini.com.br (L.B.A.); frederico.malta@grupopardini.com.br (F.S.V.M.); cristiane.brito@grupopardini.com.br (C.P.T.B.M.); alessandro.ferreira@grupopardini.com.br (A.C.S.F.); 2Laboratório de Biologia Integrativa, Departamento de Genética, Ecologia e Evolução, Instituto de Ciências Biológicas, Universidade Federal de Minas Gerais, Belo Horizonte 31270-901, Brazil; camargos.paulaluize@gmail.com (P.L.C.F.); dcqbioufmg@gmail.com (D.C.Q.); ferreira_jgg@yahoo.com (J.G.G.F.); renanpedra@gmail.com (R.P.S.); 3Departamento de Genética, Instituto de Biologia, Universidade Federal do Rio de Janeiro, Rio de Janeiro 21941-971, Brazil; filipe.moreira@biologia.ufrj.br

**Keywords:** SARS-CoV-2, variants, epidemiologic surveillance, COVID-19, transmission, symptoms

## Abstract

Coronavirus disease 2019 (COVID-19) pandemic has caused immeasurable impacts on the health and socioeconomic system. The real-time identification and characterization of new Variants of Concern (VOCs) are critical to comprehend its emergence and spread worldwide. In this sense, we carried out a national epidemiological surveillance program in Brazil from April to October 2021. Genotyping by reverse transcription-quantitative polymerase chain reaction (RT-qPCR) and sequencing were performed to monitor the dynamics and dissemination of VOCs in samples from 15 federative units. Delta VOC was first detected on June 2021 and took sixteen weeks to replace Gamma. To assess the transmissibility potential of Gamma and Delta VOCs, we studied the dynamics of RT-qPCR cycle threshold (Ct) score in the dominance period of each variant. The data suggest that Delta VOC has a higher transmission rate than Gamma VOC. We also compared relevant symptom patterns in individuals infected with both VOCs. The Delta-infected subjects were less likely to have low oxygen saturation or fatigue, altered results on chest computed tomography, and a propensity for altered X-rays. Altogether, we described the replacement of Gamma by Delta, Delta enhanced transmissibility, and differences in symptom presentation.

## 1. Introduction

According to the latest data from the World Health Organization (WHO), there are currently more than 450 million confirmed cases and approximately six million deaths caused by severe acute respiratory syndrome coronavirus 2 (SARS-CoV-2), defined as coronavirus disease 2019 (COVID-19) pandemic [1]. Viruses from the Coronaviridae family, such as SARS-CoV-2, are viruses enveloped that present spicules projected from its surface with the appearance of small crowns, giving the name of Coronavirus. They have large genomes for RNA viruses (26 to 30 kb) which is possible due to the exonuclease activity of Nsp14 protein responsible for proofreading the RNA during replication, which maintains genome stability and a mutation rate of ∼10^−6^ mutations/site/cycle [2,3,4]. Over time, the emergence and fixation of mutations resulted in new SARS-CoV-2 variants leading to adaptation of the new coronavirus to human hosts and vaccination scenario [5,6].

Experience accumulated in preventing and treating COVID-19 through virological, immunological, epidemiological, and clinical investigations have provided relevant knowledge for confronting the pandemic. However, the emergence of new fast-spreading variants of SARS-CoV-2 has caused great concern about drug and vaccine development [7]. Currently, the WHO designated five Variants of Concern (VOCs) [8], including Alpha (B.1.1.7), Beta (B.1.351), Gamma (P.1), Delta (B.1.617.2), and more recently, Omicron (B.1.1.529) [9]. These variants present mutations in Spike protein, especially in the Receptor Binding Domain (RBD) region, that have been shown to reduce recognition by neutralizing antibodies in vitro. Some of these variants have already been associated with relevant clinical-epidemiological changes and a major impact on global public health, such as increased viral transmissibility and virulence [6,10]. In this sense, identifying and characterizing emerging variants that may threaten public health constitute a critical piece of epidemiological surveillance.

Epidemiological surveillance comprises the systematic collection, analysis, and interpretation of health data that helps target strategies to combat crises caused by infectious diseases [11,12], such as the COVID-19 pandemic. Thus, SARS-CoV-2 epidemiological surveillance initiatives have enabled rapid knowledge shared between different scientific communities worldwide. These initiatives allowed the establishment of specific SARS-CoV-2 databases [13] that accelerated vaccine development [14]. Several regional initiatives have also emerged in Brazil to collect information associated with the COVID-19 pandemic [10,15,16,17]. However, due to its enormous territorial extension, a national unified genomic surveillance system is urgent to expand and connect the regional data in real-time. In this context, this work aims to present the results of SARS-CoV-2 variants monitoring in the Brazilian territory to contribute to the national epidemiological surveillance system. Our analysis explored SARS-CoV-2 variants circulating in Brazil, transmissibility potential, and symptomatology from April to October 2021.

## 2. Materials and Methods

### 2.1. Study Design

The study was divided into three main topics: (i) Genomic surveillance by reverse transcription-quantitative polymerase chain reaction (RT-qPCR) genotyping analysis and whole-genome sequencing; (ii) transmissibility analysis between Gamma and Delta VOCs by measuring RT-qPCR cycle threshold (Ct) score and; (iii) meta-analysis of the symptomatology dynamics between Gamma and Delta VOCs. The samples were collected from 15 Brazilian capitals: Belém, Belo Horizonte, Boa Vista, Brasília, Campo Grande, Fortaleza, Goiânia, Macapá, Manaus, Palmas, Porto Alegre, Porto Velho, Rio de Janeiro, Salvador, and São Paulo from April to October 2021. The Research Ethics Committee approved this study (CAAE-33202820.7.1001.5348). The authorization allows access to epidemiological and viral data with exemption to the consent form in samples from regular viral diagnosis.

### 2.2. Genomic Surveillance by RT-qPCR Genotyping Analysis and Whole-Genome Sequencing

#### 2.2.1. Genotyping Analysis of SARS-CoV-2 Variants by RT-qPCR

Nasopharyngeal swab samples were randomly selected between 20 April and 31 October 2021. RT-qPCR confirmed the SARS-CoV-2 diagnosis in Hermes Pardini Institute, a prominent Brazilian diagnostic company that performs COVID-19 tests across all 27 Brazilian federative units. We included positive samples for SARS-CoV-2 with a Ct ≤ 30. The total RNA was extracted by KingFisher Flex System instrument (Thermo Fisher, Waltham, MA, USA) using the MagMAX Viral/Pathogen Nucleic Acid Isolation Kit (Thermo Fisher, Waltham, MA, USA) according to the manufacturer’s instructions. The genotyping of SARS-CoV-2 was performed using TaqMan SARS-CoV-2 Mutation Panel (Thermo Fisher, Waltham, MA, USA) with specific primers and probes targeting the VOCs defining mutations, according to frequencies of variants in the evaluated time. We selected a set of seven non-synonymous viral spike protein mutations: K417T (A22812C), K417N (G22814T), L452R (T22917C), E484K (G23012A), E484Q (G23013C), N501Y (A23063T), and P681R (C23604G). The assays were performed by real-time quantitative PCR with the iTaq Universal Probes One-Step kit (Bio-Rad, Hercules, CA, USA). Each reaction was performed using 4.5 μL of RNA, five μL of iTaq universal probes reaction mix (2×), 0.25 μL of iScript advanced reverse transcriptase, and 0.25 μL of specific SNP probe (40×; TaqMan SARS-CoV-2 Mutation Panel) in a final volume of 10 μL. The amplification/genotyping reaction followed the cycling conditions: 50 °C for 10 min; 95 °C for 3 min; 45 cycles of 95 °C for 15 s and 60 °C for 1 min; 60 °C for 1 min. The results were plotted on moving average graphs using the ggplot2 [18] and zoo [19] packages of the R program (R version 4.1.1; R Foundation for Statistical Computing, Vienna, Austria). Datasets and codes used are available on Appendix A.

#### 2.2.2. SARS-CoV-2 Genome Sequencing

We conducted whole-genome SARS-CoV-2 sequencing in all positive samples with Ct ≤ 30 showing an unexpected spike mutation profile on the RT-qPCR genotyping. Viral RNA was amplificated by RT-qPCR using the QIAseq SARS-CoV-2 Primer Panel V1 (QIAGEN, Hilden, Germany) and ARTIC V3 multiplex primer set as previously described [16]. The library was constructed using the QIAseq FX DNA Library Prep kit (QIAGEN, Hilden, Germany), and sequencing was performed on the MiSeq platform (Illumina, San Diego, CA, USA) with v3 cartridges (600 cycles) following the manufacturer’s instructions. For each sample processing step (cDNA synthesis, viral genome amplification, and library preparation), negative controls were included in each batch of samples. All consensus genome sequences characterized in this study have been deposited on GISAID and are publicly available (Appendix A). 

#### 2.2.3. Viral Genome Assembly

The pipeline for sequencing data processing was performed as previously described [17], which includes: (i) Filtering and trimming reads and adapters using fastp v.0.20.1 [20]; (ii) mapping the sequences with the reference genome (NCBI RefSeq SARS-CoV-2; accession number: NC_045512) with Bowtie2 v2.4.2 [21]; (iii) sorting and indexing the mapped files with SAMtools v1.12 [22]; (iv) variant calling and consensus genome inference using the BCFtools v1.12; and (v) mask sites with depth less than 10× with BEDtools v2.30.0. Sequences with less than 70% coverage were removed from the analysis [23].

#### 2.2.4. Lineage Classification and Phylogenetic Analysis

The consensus genomes were classified in Pango lineages using pangolin tool v3.1.17 [24] and NextClade web application v.1.7.0 [25]. Phylogenetic analysis was performed to confirm the classification and contextualize the new sequences generated in our study. We constructed a reference dataset (*n =* 281) with representative genomes of SARS-CoV-2 available in the GISAID EpiCoV database [13] that comprised genomes generated in this study (*n =* 69) and Brazilian references (*n =* 212), SARS-CoV-2 sequences deposited in the period of April to October 2021 (acknowledgments is available in Appendix A). The dataset was aligned using MAFFT v7.480 [26], and the phylogenetic inference was performed with IQTREE [27] with the maximum likelihood method under the GTR + F + I + G4 nucleotide substitution model [28,29]. The support value of the branches was evaluated using the Shimoidara-Hasegawa-like approximate likelihood ratio (SH-aLRT) test with 1000 replicates. Dataset and script are available on Appendix A.

### 2.3. Transmissibility Analysis by Measuring Ct Values

To assess the transmissibility potential of SARS-CoV-2 variants and investigate the dominance relationship between Gamma and Delta VOCs, we studied the dynamics of RT-qPCR Ct score in samples from Brazilian capitals. RT-qPCR tests provide semi-quantitative results in the form of Ct values, which are inversely correlated with log10 viral loads and have been used to establish differences in transmissibility among viral lineages [15,17,30].

Ct data were evaluated from 61,815 patients who had positive RT-qPCR results for SARS-CoV-2 RNA, collected in 15 Brazilian capitals between 1 April and 31 October 2021. The Ct values were obtained from the amplification of three distinct viral targets (N, ORF1ab, and S genes of SARS-CoV-2), in addition to internal process control used here as a virus-independent variable (MS2), with the TaqPath COVID-19 CE-IVD RT-PCR kit (Thermo Fisher, Waltham, MA, USA) according to manufacture’s instructions. Samples were anonymized and categorized into Gamma or Delta groups when a given lineage exhibited frequency above 90% at the time of collection to estimate differences in the distribution of Ct values in periods dominated by different VOCs. The dominance period was defined for each capital: Belém (Gamma: 1 April 2021–16 June 2021; Delta: 1 October 2021–31 October 2021), Belo Horizonte (Gamma: 4 January 2021–31 July 2021; Delta: 1 September 2021–31 October 2021), Brasília (Gamma: 4 January 2021–15 June 2021; Delta: 1 September 2021–31 October 2021), Fortaleza (Gamma: 4 January 2021–15 July 2021; Delta: 1 October 2021–31 October 2021), Goiânia (Gamma: 4 January 2021–15 July 2021; Delta: 16 September 2021–31 October 2021), Palmas (Gamma: 4 January 2021–15 June 2021; Delta: 16 September 2021–31 October 2021), Porto Alegre (Gamma:4 January 2021–15 July 2021; Delta: 1 September 2021–31 October 2021), Porto Velho (Gamma: 4 January 2021–15 June 2021; Delta: 16 October 2021–31 October 2021), Rio de Janeiro (Gamma: 4 January 2021–15 June 2021; Delta: 16 August 2021–31 October 2021), and São Paulo (Gamma: 4 January 2021–15 July 2021; Delta: 16 September 2021–31 October 2021). Data from periods with intermediate frequency and data from capitals in which the Delta VOC did not reach the established dominance cut-off were not analyzed. The lineage effect on the Ct score was estimated at the national level and for each capital using a linear regression model, and results were plotted on graphs using the ggplot2 [18] packages of the R software (R version 4.1.1; R Foundation for Statistical Computing, Vienna, Austria). Dataset and code used are available on Appendix A. 

### 2.4. Symptomatology Meta-Analysis

Data available in the Severe Acute Respiratory Syndrome (SRAG) database (https://opendatasus.saude.gov.br/dataset/bd-srag-2021, accessed on 11 November 2021) were explored to assess patterns of relevant symptoms from 1 April to 31 October 2021. We selected 9814 registries that simultaneously met the following criteria: (i) samples from unvaccinated individuals, (ii) confirmation of COVID-19 diagnosis by RT-qPCR, and (iii) complete registration information. Records were collected from 15 Brazilian capitals, and the data were divided into two main groups Gamma (control) and Delta (case) groups. Each group was composed of patients’ clinical data corresponding to the period in which the variant represented at least 90% of the other variants in the capital. We were able to retrieve the following clinical data: fever, cough, sore throat, O_2_ saturation, fatigue, dyspnea, respiratory distress, diarrhea, vomiting, abdominal pain, loss of smell, loss of taste, need for ventilatory support, altered computed tomography of chest result, altered X-ray of chest result, intensive care unit (ICU) admission, and death. The meta-analysis (random and common effects) and the heterogeneity among capitals were calculated using the meta-package in R software (R version 4.1.1; R Foundation for Statistical Computing, Vienna, Austria). Dataset and code used are available on Appendix A.

## 3. Results

### 3.1. Change in the Dominance Profile of SARS-CoV-2 Variants in Brazil Territory

According to the availability and the pre-established criteria, 7549 samples were evaluated from 15 different Brazilian capitals between April and October 2021. Belo Horizonte was the capital with largest number of samples in this study (*n* = 1820), followed by Rio de Janeiro (*n* = 1097), São Paulo (*n* = 1093), Porto Alegre (*n* = 759), Brasília (*n* = 725), Goiânia (*n* = 471), Fortaleza (*n* = 401), Belém (*n* = 297), Palmas (*n* = 288), Campo Grande (*n* = 198), Porto Velho (*n* = 155), Salvador (*n* = 70), Macapá (*n* = 65), Manaus (*n* = 60), and Boa Vista (*n* = 50). Our sampling represented all regions of Brazil, covering 55.56% (15/27) of Brazilian State Capitals (Figure 1A). The sample representativeness was assessed with a cartogram (Figure 1B), which showed that the Midwest and Southeast regions had the highest sampling rates (number of samples/population size of each region) compared to the other areas studied.

According to our data, between April and July 2021, Gamma VOC was the most prevalent variant among all evaluated capitals (Figure 2A). On 1 June 2021 (epidemiological week 22), the Delta VOC was detected for the first time in Rio de Janeiro, followed by Brasília (Midwest region), São Paulo, and Belo Horizonte (both Southeast region) (Figure 2B). Subsequently, at least one case was observed in all five regions of Brazil. The spread of Delta VOC promoted a change in the variant’s dominance profile: the case number of Gamma VOC decreases in contrast to the Delta VOC expansion (Figure 2C,D). In the second half of August, Delta VOC became prevalent, comprising more than 60% of coexisting variants in Rio de Janeiro (95.31%; 61/64), São Paulo (87.67%; 64/73), Belo Horizonte (82.14%; 69/84), Porto Alegre (75.76%; 25/33), Fortaleza (63.63%; 7/11), and Goiânia (60%; 27/45). In October 2021, the Delta VOC frequency reached at least 90% of cases in Belém and Fortaleza, while in Belo Horizonte, Brasília, Goiânia, Macapá, Manaus, Palmas, Porto Alegre, Porto Velho, Rio de Janeiro, Salvador, and São Paulo reached at 99% (Figure 2D).

Besides the Gamma and Delta VOCs, other lineages were also detected but with a low frequency (Figure 2A). The Alpha VOC was identified with an average frequency of 1.76% in April 2021, only in Belo Horizonte, Brasília, and São Paulo. This lineage continued to circulate between these capitals from May to July 2021, but the frequency did not exceed 8%. In August 2021, the Alpha VOC was also identified in the capitals Goiânia, Palmas, and Porto Alegre, with an average frequency of 1.58%. In the following month, there was only one case of the Alpha VOC in Porto Alegre (0.55%), and it was not detected in any of the capitals in October (Appendix A). In parallel, we identified the Zeta lineage only in the first three months of the study, corresponding to April, May, and June 2021. Its maximum frequency was 7.69% (1/13) in the Palmas capital in April. (Appendix A). 

During the study, 2% of samples showed a different genotypic profile from the expected mutations panel, indicating the possibility of new variants not yet described. Thus, these different profiles were classified as “others” (Appendix A) and submitted to SARS-CoV-2 whole-genome sequencing.

### 3.2. Sequencing Metrics, Classification, and Phylogeny

In total, 69 samples with a non-characteristic genotypic profile by RT-qPCR genotyping were sequenced and characterized in this study. Median genome coverage of 95.71% (73.39 to 99.82%) and sequencing depth of 944.4×–2508.8× (median: 957.5×). The sequences, initially analyzed by Pangolin and NextClade tools, were classified as Gamma (P.1) (91.30%; 63/69), Alpha (B.1.1.7) (1.45%; 1/69), P.4 (1.45%; 1/69), and Delta (B.1.617.2) (5.79%; 4/69). Sequencing metrics and the pangolin/nextclade classification are available in Appendix A. We performed a maximum likelihood phylogeny to corroborate the data found in Pangolin and Nextclade classification. Our phylogenetic analysis confirmed that the novel genomes were correctly classified among VOCs Alpha, Gamma, Delta, and variant P.4 (Figure 3).

### 3.3. Delta VOC Induces a Decay in Ct Values Compared with Gamma VOC

To explore the dominance dynamics among the main SARS-CoV-2 VOCs in Brazil and to verify whether transmissibility was associated with induction of higher viral load in the upper respiratory tract, we evaluated the RT-qPCR Ct values in periods dominated by Gamma or Delta VOCs. In this sense, the dataset was categorized as Gamma group (*n* = 40,845) or Delta group (*n* = 7539), depending on which variant was dominant. The period of dominance was arbitrarily defined for each variant displaying frequency above 90%. Samples from periods in which there was no dominant VOC were excluded.

The period marked by the increasing frequency of Delta VOC is associated with lower median Ct values than the Gamma-dominated period. Median Ct values for the Gamma Period: 17.12 (N), 16.96 (ORF1ab), 17.23 (S), 25.63 (MS2). Median Ct values for the Delta period: 16.80 (N), 16.61 (ORF1ab), 16.53 (S), 25.25 (MS2). Comparative linear regression analysis at national level revealed a statistically significant negative correlation between Delta VOC and Ct value for the three viral targets analyzed (N: *p* < 0.001, β = −0.853 ± 0.075; ORF1ab: *p* < 0.001, β = −0.964 ± 0.076; S: *p* < 0.001, β = −1.258 ± 0.075) (Figure 4A–C). A small effect was observed for the control target gene (MS2: *p* < 0.001, β = −0.387 ± 0.039) due to a large number of analyzed data (Figure 4D). However, we observed a stronger correlation between the virus-related genes (N, S, and ORF1ab) and the effects of Cts values presenting in Delta infections. The data suggested Ct value reduction associated with Delta frequency increase. Therefore, subjects infected with Delta VOC presented higher viral loads in the upper respiratory tract at most Brazilian capitals (Appendix A).

### 3.4. Effect of SARS-CoV-2 Variants on Symptomatology and Clinical Data

Meta-analysis of the individual results by each capital using the extracted data from the Brazilian SRAG database indicated that individuals in the Delta group were less likely to have low oxygen saturation (Odds Ratio—OR: 0.63; 95% CI: 0.47–0.84) or fatigue (OR: 0.65; 95% CI: 0.48–0.88) than those present in the Gamma group (Figure 5A,B). There was no statistical difference between the two groups for the other symptoms such as fever, cough, respiratory discomfort, sore throat, dyspnea, diarrhea, vomiting, abdominal pain, loss of taste, and loss of smell (Appendix A). Regarding clinical examinations, individuals in the Delta group were also less likely to have altered results on chest computed tomography (OR: 0.17; 95% CI: 0.04–0.71) (Figure 5C) and a propensity for altered x-rays when compared to patients in the group of previously circulating lineage (Appendix A). In addition, an increased chance of being admitted to intensive care was also observed in the Delta variant group (OR: 1.38; 95% CI: 1.05–1.82) (Figure 5D), but no increase for mechanical ventilation or death was detected (Appendix A).

## 4. Discussion

Since December 2019, the world has faced one of the most challenging infectious diseases in decades [31]. The SARS-CoV-2 has spread worldwide, causing significant socioeconomic and health impacts on a global scale [32]. In Brazil, the devastating effects of the pandemic were no different. Brazil is the largest country in Latin America, with continental dimensions that pose challenges of comparable sizes, such as the difficulty of adopting unitary policies to confront and monitor the COVID-19 pandemic [33]. These problems can be better managed with national epidemiologic surveillance programs, which provide data that allow preventive and effective public health policies. In this context, we evaluated the epidemiological dynamics of SARS-CoV-2 variants from 15 different Brazilian capitals between April and October 2021.

The rapid description of new variants is one of the advantages of epidemiological surveillance [34]. An example was the Gamma VOC identification, first detected in Amazonas in late November 2020, which contributed to the health system’s collapse in early 2021. This variant spread rapidly across the country and was responsible for the second wave of infections, turning Brazil into the epicenter of the COVID-19 in the first trimester of 2021 [10,35]. Less than two months were sufficient for Gamma VOC to replace the parental lineage and turn the predominant SARS-CoV-2 variant in Brazil [15]. In our data, Gamma VOC was predominant up to mid-July. The first Delta VOC was detected in the Rio de Janeiro capital in June. One month later, this variant was also detected in other capitals, such as Belo Horizonte, Brasília, and São Paulo. A study using random sequencing of samples from patients with COVID-19 described that community transmission of the Delta in Rio de Janeiro possibly started in June, corroborating the results found in our study [36]. Our results demonstrated that the Delta VOC had become the predominant variant in several capitals such as Rio de Janeiro, São Paulo, Belo Horizonte, Belém, Porto Alegre (considering the 90% predominance cut-off) in September 2021.

Although Gamma (P.1) and Delta (B.1.617.2) were the main strains detected in our study, the circulation of low-frequent variants, such as Alpha (B.1.1.7), Zeta (P.2), and P.4 was also identified by RT-qPCR genotyping strategy. A total of four circulating lineages were identified by genome sequencing, in which the VOC Gamma was the main one identified. This result is consistent with the period of our study and the prevalence of this variant in Brazil. Moreover, the variants identified in our study are compatible with that reported worldwide [37,38]. The phylogenetic analysis revealed great genomic diversity among the sequences by identifying five sublineages of VOC Gamma and two sublineages of VOC Delta. Approximately 80% of the sequences have non-synonymous mutations in the S gene correlated with increased transmissibility and pathogenicity, such as D614G [39] and N501Y mutations [40].

The replacement of circulating variants by Delta VOC has been well documented [41,42]. Current evidence supports that this variant is more infectious and has a higher transmissibility rate when compared with Alpha and Beta VOCs, even in individuals with natural or vaccine-induced immunity [43,44,45]. However, studies comparing the transmissibility of Delta and Gamma VOCs are limited, and the mechanisms by which these events occur are not entirely established. Thus, to verify the differences in the potential transmissibility between these variants, we evaluated the dynamics of viral loads measured from RT-qPCR Ct values. Our results confirm the hypothesis that the Delta VOC is more transmissible than the Gamma VOC and suggest that this advantage is likely due to an increase in viral load in the upper respiratory tract as described in other studies [45,46,47]. This fact can be observed through the data comparison by linear regression in the dominated periods by the different VOCs at the national level, which indicated an association between increased frequency of Delta VOC and drops in Ct values (range β = −0. 853 to −1.258; *p* < 0.001). The data also showed that random variations during the RT-qPCR diagnostics tests were detected in the analysis. A significant decay on Ct values measured for the MS2 endogenous control target was observed (β = −0.387; *p* < 0.001). At the same time, these small random fluctuations incur an effect two–three fold less than the one reported for viral targets, not being enough to mischaracterize the differences of Ct obtained for the distinct VOCs. Recent reports on the epidemiological dynamics of SARS-CoV-2 variants have shown that transmission fitness advantages shaped the evolution of viral variants over time in diverse Brazilian states [10,17,48]. Our findings agree with these studies and help explain the substitution process for Delta over Gamma VOC at the national level. Thus, considering the Ct analysis data and other works that reinforced the increased transmissibility potential of the Delta variant [49,50], we expected that the replacement time of the Gamma variant by the Delta variant would be shorter, which, however, was not observed. One hypothesis may be related to the progress of the vaccination campaign. In Brazil, when Gamma VOC emerged, the vaccination campaign had not yet started. In September 2021, when Delta VOC became prevalent, approximately 71% of the population was vaccinated with at least the first dose, while only 43% were fully vaccinated [51]. Studies have indicated that large-scale vaccination effectively decreases virus transmission and, consequently, decreases the number of cases [52]. It can be seen when comparing the number of COVID-19 cases in January (around 69,000 cases/day) and September 2021 (near 14,000 cases/day) [53]. Thus, although there was a more transmissible variant in circulation in September, the increase of cases was not observed, reinforcing the protective effect of vaccines on the population.

According to symptomatology meta-analysis, we observed a reduction of symptoms such as fatigue and oxygen saturation in the Delta group and fewer altered exams on computed tomography (CT) and X-ray. Similar results were found in a recent study that considered the changes on the Delta-variant COVID-19 children’s chest CT milder than the original strain [54]. However, our results suggested an increase in ICU admission in the Delta group compared to the Gamma group. In another study [55], no difference was observed in the proportion of death, mechanical ventilation, or ICU admission between Delta and pre-Delta patients. Nonetheless, the pre-Delta group was not well characterized concerning the variants’ predominance. Several differences in the clinical manifestations of COVID-19 have been reported, mainly related to the emergence of new variants. COVID-19 could be classified as a multisystemic disorder and not only a respiratory disease [56], so the increase in ICU admission can be justified by other systemic complications that are not evaluated in our database. Another limitation of SRAG database was the unavailability of unvaccinated hospitalized individual data in all evaluated capitals in this study, especially in the period of the Delta variant predominance, as long the vaccination program progressed.

Furthermore, another challenge of this work was the difficulty in collecting samples from all Brazilian capitals within the eligibility criteria. These barriers are commonly faced by initiatives that propose to carry out epidemiological surveillance in a country with continental proportions.

## 5. Conclusions

Our study described the replacement of Gamma by Delta VOCs, followed by increasing transmissibility and respiratory symptoms associated between April to October 2021 in Brazil.

Over two years of the COVID-19 pandemic, the disease was marked by moments of acceleration or slowing transmission, which may be directly related to the predominant VOC and the vaccination rate. Since each VOC has different virulence capabilities, the clinical signs caused by them can also differ. Therefore, we must be aware of these changes, especially about emerging variants such as the Delta variant and, more recently, the Omicron VOC. In this sense, the emergence and rapid dominance of the Omicron VOC in a short period of time after our evaluation only reinforce the importance of genomic and epidemiological vigilance initiatives as mechanisms to monitor SARS-CoV-2 variants.

Finally, the epidemiological surveillance system described here has the potential to be expanded to monitor new variants. This strategy can provide subsidies to health authorities to implement effective control strategies to prevent COVID-19 transmission.

## Figures and Tables

**Figure 1 viruses-14-00847-f001:**
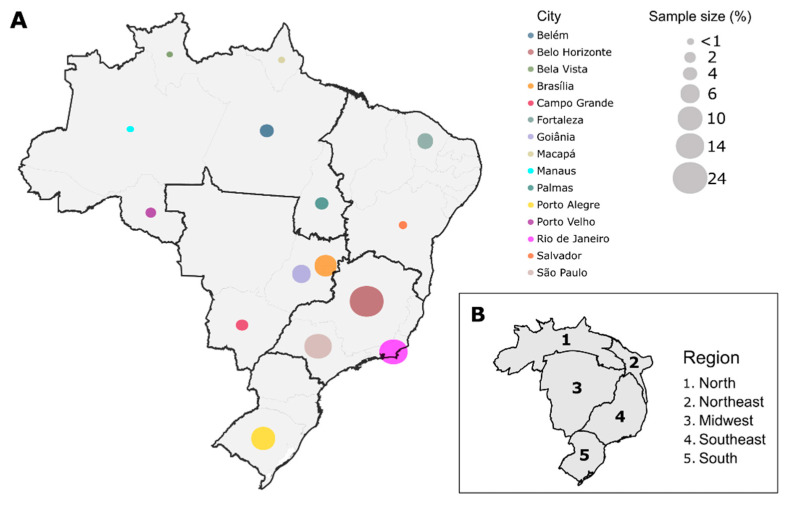
Schematic representation of study sampling. (**A**) Samples from 15 different capitals of Brazil, representing the five regions (North, Northeast, Midwest, Southeast, and South) were included in our study. In total, 7549 samples were analyzed and the proportion of samples for each capital is represented by the size of the colored circles. Each color represents a different capital; (**B**) Cartogram of the sampling representativeness according to the population size of each Brazilian region.

**Figure 2 viruses-14-00847-f002:**
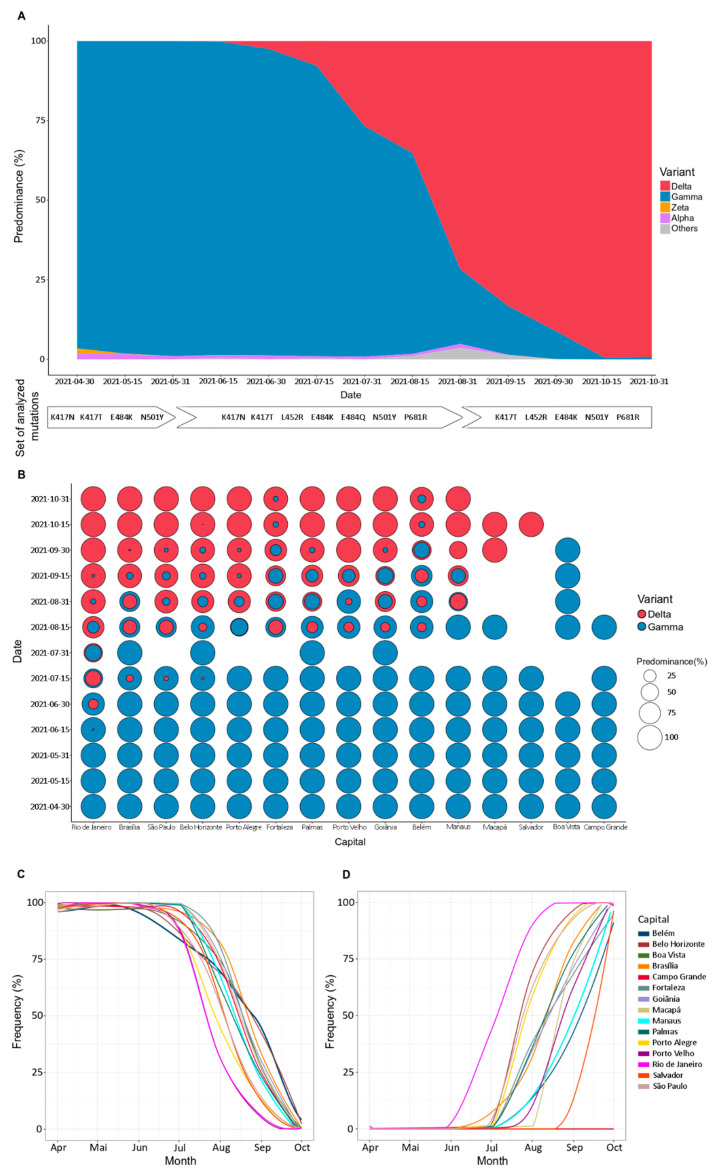
Most prevalent severe acute respiratory syndrome coronavirus 2 (SARS-CoV-2) variants detected in Brazil by genotyping analysis. (**A**) Gamma (blue) and Delta (red) were the main variants detected in the period evaluated. In the first months of monitoring, Gamma was predominant but gradually replaced by Delta. Other variants, such as Alpha (purple) and Zeta (yellow), were also detected at low frequency. All samples that could not be classified by genotyping were labeled as “Others” (gray) and selected for sequencing analysis. White tags indicate the analyzed mutations set, considering the expected frequencies of the variants in the evaluated time; (**B**) Each capital presented a slope of increase in new cases by Delta while Gamma cases decay. Rio de Janeiro was the first capital where Delta was detected (early June) and replaced Gamma entirely (last September). In other capitals, such as Fortaleza and Belém, the emergence of Delta was late, and the complete replacement of Gamma by Delta was not observed in the period evaluated. Blank spaces indicate the absence of samples in a particular capital within the specified time; (**C**) Gamma was the most prevalent variant among all evaluated capitals up to July 2021; (**D**) Delta was detected for the first time in June 2021, in Rio de Janeiro. Initially, Delta spread through the Southeast and Midwest regions and, subsequently, for all Brazilian macro-regions, promoting a change in the genotypic profile until then observed, becoming predominant in the second half of August.

**Figure 3 viruses-14-00847-f003:**
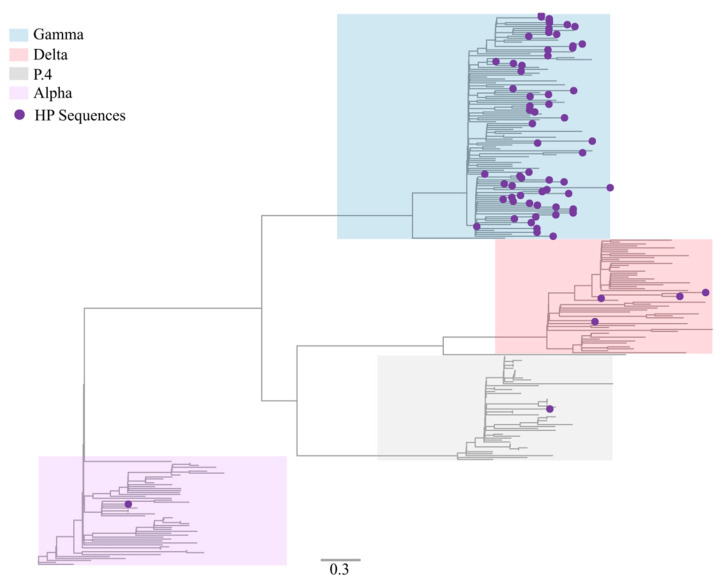
Maximum likelihood inference tree to confirm lineage classification using a reference dataset. The genomes generated in our study (*n* = 69) are highlighted with purple circles. Lineages Alpha (B.1.1.7), Gamma (P.1), Delta (B.1.617.2), and P.4 are highlighted in violet, blue, red, and gray, respectively.

**Figure 4 viruses-14-00847-f004:**
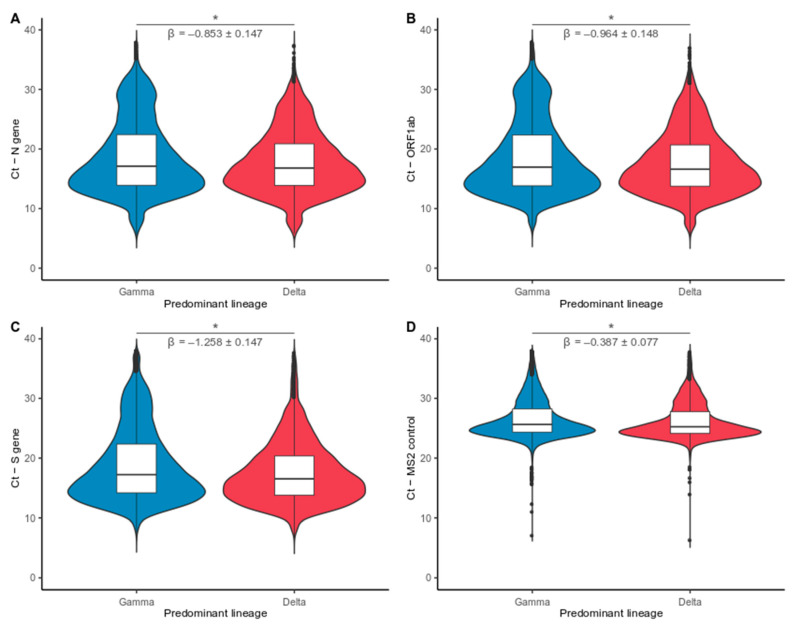
Comparative analysis of reverse transcription-quantitative polymerase chain reaction (RT-qPCR) cycle threshold (Ct) values between Gamma and Delta dominated periods. Violin plots displaying the distribution of Ct data for different Variants of Concern (VOCs) for three viral targets and internal control: (**A**) N gene; (**B**) ORF1ab; (**C**) S gene; and (**D**) MS2 control. Sample groups were input as Gamma (blue) and Delta (Red) when the variant exhibited frequency above 90%. Statistical comparison between periods denotes that Delta VOC induces higher viral loads in the upper respiratory tract than Gamma VOC infection. Internal control (MS2) had an effect two–three fold smaller on Ct decrement than reported for all viral targets. Asterisks indicate a significant statistical association between imputed viral lineages and Ct values (linear model: *p* < 0.001). * *p*  <  0.05.

**Figure 5 viruses-14-00847-f005:**
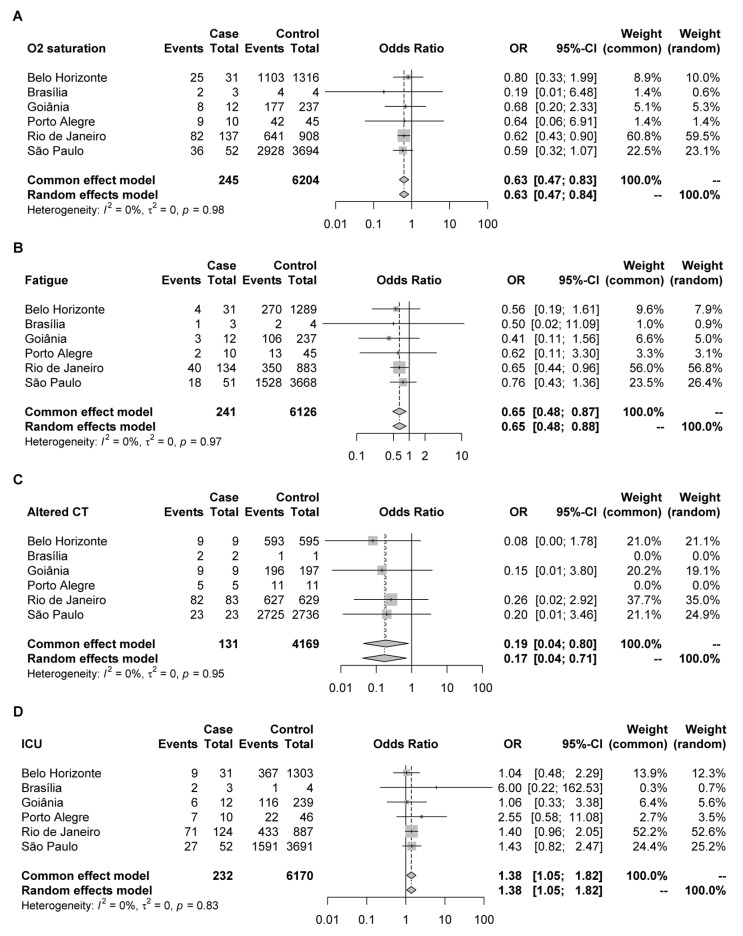
Symptoms/clinical tests occurrence in Delta (case) and Gamma (control) groups in Coronavirus disease 2019 (COVID-19) patients from six different capitals of Brazil. The Forest plot describes the observations for: (**A**) O_2_ saturation; (**B**) fatigue; (**C**) altered chest computed tomography; (**D**) intensive care unit (ICU) admission.

## Data Availability

All generated genome sequences have been deposited on GISAID (IDs: EPI_ISL_ 6500240, EPI_ISL_6500295, EPI_ISL_9383777 to EPI_ISL_9383784, EPI_ISL_9383786 to EPI_ISL_9383804, and EPI_ISL_9383806 to EPI_ISL_9383846).

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
