# Peer review of "Delta Variant of SARS-CoV-2 Replacement in Brazil: A National Epidemiologic Surveillance Program"

_viruses, 2022, doi:10.3390/v14050847_

Round 1

Reviewer 1 Report

Dear Authors 

The abstract of the paper and title is fine. Keywords of the abstract are not good try to add proper MeSH terms. 

In the introduction, the authors can improve the gaps by reading this paper and using this information.

1) Khurshid, Z.; Asiri, F.Y.I.; Al Wadaani, H. Human Saliva: Non-Invasive Fluid for Detecting Novel Coronavirus (2019-nCoV). Int. J. Environ. Res. Public Health 202017, 2225.

2) Yadalam, Pradeep Kumar, et al. "Assessing the therapeutic potential of angomelatine, ramelteon, and melatonin against SARS-Cov-2." Saudi Journal of Biological Sciences (2022).

From lines 64 to 71, need English improvement.

Methods are well written but have few language issues. Need to revise wherever possible. 

What is the significance of this study? please add before the conclusion heading. 

Author Response

Dear Reviewer 1,

Your comments are in bold letters, followed by our responses. All the revisions in the manuscript are marked up using the "Track Changes" function.

1-The abstract of the paper and title is fine. Keywords of the abstract are not good try to add proper MeSH terms.

  • Thank you for your review and suggestion. As requested, we have made changes in the Keywords section of the revised manuscript as follows: "Keywords: SARS-CoV-2, variants, COVID-19, epidemiologic surveillance, transmission, symptoms."

2-In the introduction, the authors can improve the gaps by reading this paper and using this information.

- Khurshid, Z.; Asiri, F.Y.I.; Al Wadaani, H. Human Saliva: Non-Invasive Fluid for Detecting Novel Coronavirus (2019-nCoV). Int. J. Environ. Res. Public Health 2020, 17, 2225.

- Yadalam, Pradeep Kumar, et al. "Assessing the therapeutic potential of angomelatine, ramelteon, and melatonin against SARS-Cov-2." Saudi Journal of Biological Sciences (2022).

  • Thanks to the reviewer for the references suggestion. We have read these studies and summarized the main points of each suggested work:
    • Khurshid et al. described non-invasive methods for detecting SARS-CoV-2, including human saliva.
    • Yadalam et al. conducted an in-silico analysis to investigate if melatonin and related drugs, namely ramelteon and agomelatine, could be used as antiviral agents in SARS-CoV-2 infection.

Although being interesting works, we understand that they are not in the scope of our study. For this reason, we prefer not to include these references.

3-From lines 64 to 71, need English improvement.

  • Thanks again to the reviewer's observation, and we reformulated the text manuscript.

4-Methods are well written but have few language issues. Need to revise wherever possible.  

  • Thank you. We sent this work for English review.

5-What is the significance of this study? please add before the conclusion heading.  

  • Thanks again to the reviewer's observation, and we reformulated the Conclusion section.

Reviewer 2 Report

This study mainly investigated a national epidemiological surveillance and the characterization of SARS-CoV-2 VOCs in Brazil. There is no data about most recently Omicron VOC, and thus the significance of this work is greatly reduced. In addition,numerous sentence throughout the manuscript is hard to be understood, and the quality of English writing needs substantial improvement.

Author Response

Dear Reviewer 2,

Your comments are in bold letters, followed by our responses. All the revisions in the manuscript are marked up using the "Track Changes" function.

1-This study mainly investigated a national epidemiological surveillance and the characterization of SARS-CoV-2 VOCs in Brazil. There is no data about most recently Omicron VOC, and thus the significance of this work is greatly reduced.

  • We thank the reviewer for the comments. However, the purpose of this study was to monitor SARS-CoV-2 variants between April and October 2021, which was markedly the transition period between Gamma to Delta VOCs predominance in Brazil. The emergence and rapid dominance of the Omicron Variant in a short period after our evaluation period only reinforce the importance of initiatives like our work in establishing mechanisms to monitor SARS-CoV-2 variants in real-time. The first introduction of Omicron in Brazil, dated November 26th, 2021, is out of the scope of the current study. Considering the importance of Omicron in the epidemiological scenario of COVID-19, we added this information to the Conclusion section.

2-In addition, numerous sentence throughout the manuscript is hard to be understood, and the quality of English writing needs substantial improvement.

  • Thank you. We performed an English review in the revised manuscript.

Reviewer 3 Report

Silva et al. carried out a national epidemiological surveillance program for the new delta Variant of Concern (VOCs) using RT-qPCR. It’s very interesting study and acceptable for publication.

Author Response

Dear Reviewer 3,

Your comments are in bold letters, followed by our responses. All the revisions in the manuscript are marked up using the "Track Changes" function.

Silva et al. carried out a national epidemiological surveillance program for the new delta Variant of Concern (VOCs) using RT-qPCR. It's very interesting study and acceptable for publication.

  • We acknowledge Reviewer 3 for this comment.

Reviewer 4 Report

In this study Silva and colleagues present the data from a national epidemiological surveillance program to examine the prevalence of VOCs in Brazil over a 6-month period from April to October 2021. They report that over this period Delta supplanted Gamma as the dominant variant and that former was associated with higher transmission rates. They also show an alteration in clinical symptoms in patients infected with Delta compared to Gamma.

The study is well done and data nicely presented. I have no major critiques but only minor points for correcting some confusion.

Minor points:

One of the key findings of this study is that Delta VOC induces higher viral loads in patients than Gamma and hence is more transmissible. This was done by comparing the Ct values during periods where one VOC dominated over another.

  1. Did the authors consider doing a viral titre:RNA ratio, even on a subset of samples, as a measure of infectivity rather than just relying on genomic RNA. This has been done recently by Despres etal (https://www.medrxiv.org/content/10.1101/2021.09.07.21263229v1.full.pdf)?
  2. Could the authors state the time period for each variant that was taken into consideration in this analysis?
  3. In Fig 2A, the median Ct values for all three gene targets look very similar between Gamma and Delta although the differences were found to be statistically significant. For sake of clarity could the authors provide median Ct values for each gene target for both Gamma and Delta.
  4. Were the variables such as age, sex, days post onset of symptoms taken into account while doing the regression analysis?
  5. The reference Moreira et al (Line 483) appears twice in the reference list.
  6. Overall, the manuscript would benefit from some tighter editing especially for language and grammar.

Author Response

Dear Reviewer 4,

Your comments are in bold letters, followed by our responses. All the revisions in the manuscript are marked up using the "Track Changes" function.

In this study Silva and colleagues present the data from a national epidemiological surveillance program to examine the prevalence of VOCs in Brazil over a 6-month period from April to October 2021. They report that over this period Delta supplanted Gamma as the dominant variant and that former was associated with higher transmission rates. They also show an alteration in clinical symptoms in patients infected with Delta compared to Gamma.

The study is well done and data nicely presented. I have no major critiques but only minor points for correcting some confusion.

 One of the key findings of this study is that Delta VOC induces higher viral loads in patients than Gamma and hence is more transmissible. This was done by comparing the Ct values during periods where one VOC dominated over another.

  1-Did the authors consider doing a viral titre: RNA ratio, even on a subset of samples, as a measure of infectivity rather than just relying on genomic RNA. This has been done recently by Despres et. al.

(https://www.medrxiv.org/content/10.1101/2021.09.07.21263229v1.full.pdf)?

  • Although we would like to conduct virus titration experiments, they require Biosafety Level 3 Laboratories to which we don't have access. Therefore we were not able to perform them. Indeed, the correlation among Ct values, patient viral load, and virus transmission rates have been reported before, and we have added references supporting that relation (Refs 11 and 34):

Hay, J.A.; Kennedy-Shaffer, L.; Kanjilal, S.; Lennon, N.J.; Gabriel, S.B.; Lipsitch, M.; Mina, M.J. Estimating epidemiologic dynamics from cross-sectional viral load distributions. Science (80-. ). 2021, 373, doi:10.1126/science.abh0635.

Faria, N.R.; Mellan, T.A.; Whittaker, C.; Claro, I.M.; Candido, D. da S.; Mishra, S.; Crispim, M.A.E.; Sales, F.C.S.; Hawryluk, I.; McCrone, J.T.; et al. Genomics and epidemiology of the P.1 SARS-CoV-2 lineage in Manaus, Brazil. Science (80-. ). 2021, 372, 815–821, doi:10.1126/science.abh2644.

2-Could the authors state the time period for each variant that was taken into consideration in this analysis?

  • The period for each variant that was considered has been added to the "Materials and Methods" section (2.3 topic).

3-In Fig 2A, the median Ct values for all three gene targets look very similar between Gamma and Delta although the differences were found to be statistically significant. For sake of clarity could the authors provide median Ct values for each gene target for both Gamma and Delta.

  • The median Ct values for all three gene targets for each variant were added to the "Results" section.

4-Were the variables such as age, sex, days post onset of symptoms taken into account while doing the regression analysis?

  • We did not perform a regression analysis to explore VOC effects on clinical features. The methodology used was the meta-analysis computing the odds ratio for each symptom or clinical outcome between the two populations without making corrections for any variable.

5-The reference Moreira et al (Line 483) appears twice in the reference list.

  • We found the error on lines 513 and 519. Thanks for the review. We corrected it in the "reference" section and manuscript.

6- Overall, the manuscript would benefit from some tighter editing, especially for language and grammar.

  • We sent this work for English proofreading.

Round 2

Reviewer 1 Report

Dear Authors 

Great. Well done for revision. I have a serious advice before thi spaper goes for pubication. Before conclusion heading add few lines on the limitations observed during conducting this study and writing this work. 

Regarding conclusion is improved but no new message is written from this work. It will be great if authors improve it. 

Author Response

Dear Reviewer 1,

Your comments are in bold letters, followed by our responses. All the revisions in the manuscript are marked up using the "Track Changes" function.

Great. Well done for revision. I have a serious advice before this paper goes for pubication. Before conclusion heading add few lines on the limitations observed during conducting this study and writing this work.

Thank you for your consideration. We have added a text before the conclusion highlighting some limitations of the study.

Regarding conclusion is improved but no new message is written from this work. It will be great if authors improve it.

Thank you for your feedback. However, we understand that this work brings a new important message describing the substitution of Gamma by Delta on a national scale (Brazil), adding information about the potential for transmissibility and symptoms. We did not find studies that have included and analyzed these factors simultaneously.

Reviewer 2 Report

This version has been greatly improved. Thanks for authors' effort.

Author Response

Dear Reviewer 2

Thank you for helping us to improve this work.